# OpenReview forum: "ExpGuard: LLM Content Moderation in Specialized Domains"
_ICLR.cc/2026/Conference — ICLR 2026 Poster_

### Official Review · Reviewer_YTqP · 2025-10-30

**Soundness:** 3
**Presentation:** 3
**Contribution:** 3
**Rating:** 8
**Confidence:** 4

**Summary:**

This work introduces ExpGuard, a Guard model trained on ExpGuardMix, which consists of 58,928 prompts, including prompts from the specialized domains of Finance, Medicine, and Law. This dataset addresses the lack of domain-specific prompts for training guardrail systems. The Authors describe in high detail the creation of ExpGuardMix with a novel construction pipeline that utilized domain-specific terminology. In addition to specialized examples, ExpGuardMix incorporates real-world user-LLM conversations and human-written prompts, making ExpGuard models useful outside the specialized domains. ExpGuard is evaluated against other state-of-the-art guardrails, highlighting its advantage over other available solutions, both open-source and closed API models.

**Strengths:**

The creation pipeline of ExpGuard is novel and addresses a crucial problem when trying to apply guardrails to domain-specific applications, such as medical. Especially, the usage of domain-specific terms extracted from Wikipedia ensures that generated prompts utilize language specific to the domains of interest. The dataset creation process is highly selective, ensuring that only high-quality data is able to pass the filtering pipeline. Finally, the Authors evaluate the usefulness of ExpGuardMix by training ExpGuard and comparing it to other state-of-the-art guardrails. These results show a significant advantage of ExpGuard over other models on specialized domains, as on average ExpGuard achieves an increase of ~10% over the next best guardrail. Another experiments show how ExpGuard performs on other well-known harmfulness datasets (e.g., ToxiChat, XSTest WildGuardMix). These results again validate that ExpGuard is a highly effective guardrail, as, on average is the best and second-best model on prompt and response classification tasks accordingly. I believe that this work is of high significance to the AI Safety community.

**Weaknesses:**

* ExpGuardMix seems not to address jailbreak-like prompts, which could be a limitation for ExpGuard's robustness to domain-specific jailbreaks.
* The Authors only utilized models with fewer than 8B parameters for training ExpGuard models. I would be highly interested in the scaling above 8B models.
* For the generation of responses, Mistral-7B-Instruct-v0.1 was used, which is a model with low capabilities compared to newer and larger models. I believe that the usage of a larger model would be of high benefit for the quality of the responses in the dataset. My biggest concern is with the responses generated by this Mistral model being of low quality.

**Questions:**

* Would it be beneficial to use a more capable model for the generation of compliant responses, as this early version of Mistral could hallucinate, especially for domain-specific prompts?
* Have the Authors tested how samples would be classified if "safe" and "unsafe" were done by majority voting per category? That is, a sample would be deemed unsafe if the majority of models have returned at least one exactly the same class from 13 predefined harm categories.
* Can the Authors provide rough numbers of how many prompts were discarded in each filtering phase?
* Have the Authors added jailbreak-like examples into domain-specific harmful data?
* Regarding Figure 4: Could the Authors also show the performance of those models on these prompts before adding jailbreaks?

---

> ### Author Response · Authors · 2025-11-20
> **Response to Reviewer YTqP**
>
> *We appreciate Reviewer YTqP for their thorough feedback. We hope our responses address the reviewer’s concerns.*
>
> ### **YTqP-W1 & Q4**
>
> > ExpGuardMix seems not to address jailbreak-like prompts, which could be a limitation for ExpGuard's robustness to domain-specific jailbreaks. Have the Authors added jailbreak-like examples into domain-specific harmful data?
>
> Thank you for highlighting the crucial area of jailbreak robustness. We would like to clarify that our original dataset did include 270 general jailbreak prompts (from DAN), as noted in $\S$3.1.2. However, you are absolutely correct that it lacked domain-specific jailbreaks, which is a key limitation. Following your suggestion, we generated **270 new, domain-specific jailbreak prompts** using AutoDAN-Turbo and included them in ExpGuardMix. We then retrained our model, which we call **ExpGuard+**, on this updated dataset. This new model shows a notable increase in robustness, especially against domain-specific (DS) attacks. **We updated Section 4.4 and Figure 4** with these new, much stronger results. As the table below shows, ExpGuard+ now achieves near-perfect detection on domain-specific jailbreaks.
>
> The abbreviations stand for: `Std` (Standard), `DS` (Domain-Specific), `O` (Original), `CC` (CipherChat), and `ADT` (AutoDAN-Turbo).
>
> | Model | Std-O | Std-CC | Std-ADT | DS-O | DS-CC | DS-ADT |
> | --- | --- | --- | --- | --- | --- | --- |
> | Llama-Guard3 | 97.3 | 89.4 | 75.8 | 51.0 | 65.6 | 54.3 |
> | Aegis-Guard-D | 67.3 | 74.6 | 72.6 | 62.3 | 70.0 | 70.6 |
> | Aegis-Guard-P | 58.0 | 59.3 | 62.3 | 44.2 | 46.7 | 50.5 |
> | WildGuard | 98.8 | 89.9 | 79.3 | 70.8 | 76.9 | 70.3 |
> | ExpGuard | **99.0** | 89.7 | 78.4 | 97.7 | 87.9 | 74.0 |
> | ExpGuard+ | 95.2 | **98.9** | **82.5** | **98.5** | **94.6** | **97.1** |
>
> ### **YTqP-Q5**
>
> > Regarding Figure 4: Could the Authors also show the performance of those models on these prompts before adding jailbreaks?
>
> The performance of the models on the prompts before applying jailbreak techniques is reported in the `Std-O` (Standard-Original) and `DS-O` (Domain-Specific-Original) columns in the table from our response to **W1 & Q4**. During this analysis, we observed a peculiar phenomenon. For some models, like Llama-Guard3 and Aegis-Guard, the unsafety detection rate paradoxically increases after jailbreak techniques are applied (e.g., `DS-CC` > `DS-O`). We hypothesize this is because these guardrails fail to understand the nuanced, semantic harm in the original, normal domain-specific prompts, leading to low scores. However, the jailbreak methods introduce obvious structural or stylistic artifacts. The weaker guardrails, acting as simple pattern-detectors, catch these "jailbreak-like" patterns and flag the prompt, even if they miss the original domain-specific harm. They are detecting the attack's form rather than its content, which paradoxically makes the prompt easier for them to classify as unsafe. **We added this discussion to $\S$4.4.**

---

> ### Author Response · Authors · 2025-11-20
> **Response to Reviewer YTqP - 2**
>
> ### **YTqP-W2**
>
> > I would be highly interested in the scaling above 8B models.
>
> We initially chose a 7B model as a backbone for a fair comparison against other state-of-the-art guardrail models, which are often in the 7B range. This size also prioritizes practicality and low latency, which are critical for a guardrail's real-world deployment. To test whether our proposed ExpGuardMix dataset scales effectively, we followed your suggestion and trained a larger **Qwen2.5-14B** model. The results (which **we added to Appendix E.2 as an extension of Table 12**) show a consistent performance improvement with model scale, demonstrating that our dataset effectively enhances the capabilities of larger models as well.
>
> The abbreviations stand for: `PH` (Prompt Harm), `RH` (Response Harm), `Pub` (Public), `Exp` (ExpGuardTest), `Unwgt` (Unweighted).
>
> | Model | PH-Pub F1 | PH-Exp F1 | RH-Pub F1 | RH-Exp F1 | Avg (Unwgt) |
> | --- | --- | --- | --- | --- | --- |
> | Qwen2.5-7b (ExpGuard) | 85.7 | 93.3 | 78.5 | 92.7 | 87.6 |
> | Qwen2.5-14b | **87.7** | **95.3** | **79.0** | **93.7** | **88.9** |
>
> ### **YTqP-W3 & Q1**
>
> > Would it be beneficial to use a more capable model for the generation of compliant responses, as this early version of Mistral could hallucinate, especially for domain-specific prompts?
>
> This is an insightful suggestion regarding the quality of the generated responses. We share your concern that the older Mistral-7B-v0.1 might produce lower-quality or hallucinated content compared to modern models. To investigate this, we re-generated the training responses using the stronger **Mistral-Small-3.2-24B-Instruct**. We carefully controlled the generation process to maintain the same distribution of safe and unsafe labels as the original dataset, isolating the impact of *response quality* rather than refusal rates.
>
> | **Model (Generator)** | **BeaverT** | **S-RLHF** | **Aegis2** | **HarmB** | **WG** | **Public Avg.** | Fin | Med | Leg | **ExpTest Total** |
> | --- | --- | --- | --- | --- | --- | --- | --- | --- | --- | --- |
> | Mistral-7B-v0.1 (Original) | **81.8** | 64.1 | **82.7** | **85.5** | **78.3** | **78.5** | 96.7 | 86.2 | 92.4 | 92.7 |
> | Mistral-Small-3.2-24B | 78.5 | 64.1 | 70.5 | 83.2 | 75.0 | 74.3 | **96.8** | **90.7** | **94.8** | **94.5** |
>
> The results reveal an interesting trade-off between domain specialization and general benchmarks.
>
> 1. **Domain Performance Improved:** As you hypothesized, using the stronger model improved performance on ExpGuardTest (92.7% $\to$ 94.5%). This suggests that the 24B model generates more coherent and technically precise harmful responses in specialized domains, providing a higher-quality training signal for detecting domain-specific risks.
> 2. **General Performance Decreased:** Conversely, performance on public benchmarks dropped. We attribute this to a distribution mismatch. Many existing public benchmarks (like BeaverTails) were constructed using older or smaller models. The responses from Mistral-7B likely align closer to the stylistic distribution of these benchmarks, whereas the "smarter" responses from Mistral-Small-24B may be distinct enough to slightly reduce transferability to these older test sets.
>
> Given this trade-off, we believe the original Mistral-7B configuration offers the most balanced performance for a general-purpose release, but **we included this experiment in Appendix E.4** to highlight how stronger generator models can further boost domain-specific safety.

---

> ### Author Response · Authors · 2025-11-20
> **Response to Reviewer YTqP - 3**
>
> ### **YTqP-Q2**
>
> > Have the Authors tested how samples would be classified if "safe" and "unsafe" were done by majority voting per category? That is, a sample would be deemed unsafe if the majority of models have returned at least one exactly the same class from 13 predefined harm categories.
>
> Thank you for this excellent and highly insightful question. We must apologize, as our description in Section 3.1.3 simplified the process for brevity and was not sufficiently clear. **Your suggested method—majority voting per category—is, in fact, the exact, stricter method we implemented.** We did *not* use a simple binary vote. A sample was only included in ExpGuardMix if at least two of our three LLM classifiers (Claude, Gemini, Qwen) agreed on the **exact same harm category**. To be clear: if all three models voted “unsafe” but with *different* categories (e.g., S1, S3, and S11), we considered this an ambiguous case and **the sample was discarded**. This ensures the "highly selective" process and high inter-rater reliability that you correctly identified as a strength of our work.
>
> To provide full transparency on this process, we are happy to share the consensus statistics for our domain-specific data generation (Financial, Medical, and Legal), which includes both harmful and benign prompts:
>
> | Consensus Level                          | Finance | Healthcare |  Law | Overall          |
> | :--------------------------------------- | ------: | ---------: | ---: | ---------------: |
> | 3-way Match (All 3 models agreed)        |  7,476  |     5,110  | 4,432 | 17,018 (64.3%)  |
> | 2-way Match (2 of 3 models agreed)       |  2,219  |     4,171  | 1,785 |  8,175 (30.9%)  |
> | No Match (All 3 disagreed)               |    195  |       839  |   233 |  1,267 (4.8%)   |
> | **Total Generated**                      | **9,890** | **10,120** | **6,450** | **26,460** |
>
> As this data shows, our final dataset consists of the samples with a 2-way match (30.9%) or a 3-way match (64.3%), totaling 95.2% of the generated data. The remaining 4.8% of samples, where all three models disagreed on the category, were discarded as "No Match" to ensure dataset quality and remove ambiguity. **We revised $\S$3.1.3 and Appendix A.11** much clearer to correctly reflect this rigorous methodology. Thank you for helping us improve this key point.
>
> ### **YTqP-Q3**
>
> > Can the Authors provide rough numbers of how many prompts were discarded in each filtering phase?
>
> We have compiled the exact numbers for our domain-specific data filtering process, which **we have added to Appendix A.11 and Table 13**. The table below shows the number of prompts remaining after each filtering stage, following the initial generation:
>
> | Filtering Stage                      | Finance | Healthcare |  Law | Overall |
> | :----------------------------------- | ------: | ---------: | ---: | ------: |
> | **1. Immediately after Generation**  |  9,890  |    10,120  | 6,450 | 26,460 |
> | **2. After Majority Vote**           |  9,695  |     9,281  | 6,217 | 25,193 |
> | **3. After Deduplication (Final)**   | **9,068** | **7,712** | **5,402** | **22,182** |
>
> To answer your question about the number of discarded prompts at each phase:
>
> - **Phase 1 (Majority Vote):** A total of **1,267 prompts** (26,460 - 25,193) were discarded. As we explained in our answer to Q2, these represent the 4.8% of samples where our three LLM classifiers failed to reach a 2-way consensus on the *exact category* ("No Match"), ensuring all ambiguous content was removed.
> - **Phase 2 (Deduplication):** A further **3,011 prompts** (25,193 - 22,182) were discarded during our deduplication step. This was a crucial phase where we programmatically remove near-duplicates (cosine similarity > 0.9), ensuring the final dataset's diversity.
>
> The final **22,182** high-quality, high-agreement, and unique domain-specific samples were then used to form our ExpGuardMix.

---

> > ### Comment · Reviewer_YTqP · 2025-11-24
> >
> > I appreciate the authors' clarifications. As my concerns have been met, I will maintain my initial positive score.

---

> > > ### Author Response · Authors · 2025-11-26
> > > **Thank you!**
> > >
> > > We are pleased to hear that our response has satisfactorily resolved your concerns. If there are any additional details we can provide, please do not hesitate to let us know.

---

### Official Review · Reviewer_52Br · 2025-11-01

**Soundness:** 2
**Presentation:** 3
**Contribution:** 2
**Rating:** 4
**Confidence:** 4

**Summary:**

This paper introduces EXPGUARD, a domain-specialized LLM guardrail model designed for content moderation in finance, healthcare, and law domains, where existing general-purpose safety models often fail to recognize harmful content expressed in technical jargon. The authors also present EXPGUARDMIX, a large-scale dataset of labelled samples and an expert-verified test subset, featuring harmful and benign prompts/responses crafted from domain-specific terminology. The dataset construction pipeline involves domain-term mining, prompt/response generation, and an ensemble labelling using multiple proprietary LLMs with chain-of-thought rationales. Experimental results demonstrate that EXPGUARD achieves state-of-the-art performance and maintains competitive results on general public safety datasets.

**Strengths:**

-	The data construction pipeline for combining automated term mining, GPT-based generation, ensemble CoT-reasoning labelling, and expert verification is automatic and replicable for other domains.
-	The experimental evaluation is comprehensive, spanning both specialized and public benchmarks, with clear ablation studies showing the effect of each dataset component.
-	The paper is well-structured, with clear figures illustrating data composition, pipeline, and example prompts.

**Weaknesses:**

-	Unknown domain overlaps with existing datasets like WildGuardMix. Even though this paper focuses on special domains, it is hard to tell how different the proposed dataset is from existing datasets. It is observed that other guardrail models like WildGuard and LlamaGuard can already achieve over 70% or 80% F1 scores. This suggests that EXPGUARDMIX could only serve as a supplement to the existing data. Without the quantitative measurement of the overlapped data (both benign and harmful), it is hard to judge the value of the proposed dataset.

-	Limited evaluation. All reported evaluations are F1 scores. It is equally important to see the False Positive Rate (FPR), False Negative Rate (FNR), and prediction uncertainty to analyze what types of errors the guardrail models make, especially in high-stakes domains.

-	For both domain-specific and standard AutoDAN-Turbo jailbreaks in Figure 4, the ExpGuard trained with domain-specific data achieves little improvement on the unsafety detection rate compared to WildGuard, which makes me doubt its effectiveness when faced with other stronger jailbreak attacks.

**Questions:**

-  Why fine-tune a new guardrail model from a base model? Why not just fine-tune an existing general guardrail model to keep both general and domain-specific capabilities?

---

> ### Author Response · Authors · 2025-11-20
> **Response to Reviewer 52Br**
>
> *We appreciate Reviewer 52Br for their insightful feedback. We hope our responses address the reviewer’s concerns.*
>
> ### **52Br-W1**
>
> > Without the quantitative measurement of the overlapped data (both benign and harmful), it is hard to judge the value of the proposed dataset.
>
> We agree that proving the distinctiveness of ExpGuardMix compared to existing datasets is essential to demonstrate its value beyond being a mere supplement. To address your concern quantitatively, we conducted **a semantic overlap analysis between ExpGuardMix and three major baseline datasets**: WildGuardMix (WildGuard), SALAD-Bench (MD-Judge), and BeaverTails (BeaverDam). We generated sentence-BERT embeddings for all prompts and calculate the cosine similarity between every sample in ExpGuardMix and the baseline datasets. A sample was considered "overlapping" if its maximum similarity score exceeded a high threshold of 0.9.
>
> | Baseline Dataset | Overlap Ratio (%) |
> | --- | --- |
> | WildGuardMix (WildGuard) | 5.1 |
> | SALAD-Bench (MD-Judge) | 5.7 |
> | BeaverTails (BeaverDam) | 18.6 |
>
> Most notably, **the overlap with WildGuardMix and SALAD-Bench is extremely low at 5.1% and 5.7%, respectively.** This quantitatively confirms that the vast majority (~95%) of ExpGuardMix consists of unique, domain-specific content that is not represented in current state-of-the-art guardrail datasets.
>
> The comparatively higher overlap with BeaverTails (18.6%) is a direct result of shared data lineage. As detailed in Section 3.1.2, we explicitly integrated high-quality prompts from HH-RLHF to ensure general safety. Since BeaverTails is also constructed based on the HH-RLHF dataset, this overlap is structural and expected. This confirms that ExpGuardMix successfully retains foundational safety capabilities.
>
> These results strongly support that ExpGuardMix is not just a supplement, but a distinct dataset that covers a new distribution of risks (domain-specific technical jargon) that existing datasets fail to capture. **We included this quantitative analysis in Appendix A.10** to clarify the dataset's novelty.
>
> ### **52Br-W3**
>
> > Makes me doubt its effectiveness when faced with other stronger jailbreak attacks.
>
> We thank the reviewer for the comment regarding the robustness of ExpGuard against adversarial attacks. To address your concern and demonstrate ExpGuard's effectiveness against strong adversarial attacks, we conducted additional experiments using the most recently developed jailbreak attacks **FlipAttack [1]** and **GASP [2]** on both standard and domain-specific datasets. **We incorporated these results into Section 4.4 and Figure 4.**
>
> As the table below shows, ExpGuard exhibits robustness against sophisticated jailbreak attacks, notably achieving near-perfect detection rates on FlipAttack and outperforming WildGuard by substantial margins (e.g., +22.0% on DS-FA). These results confirm that ExpGuard effectively generalizes to stronger attacks, consistently surpassing the baseline across all domain-specific attacks while maintaining competitive performance on standard benchmarks.
>
> The abbreviations stand for: `Std` (Standard), `DS` (Domain-Specific), `CC` (CipherChat), `ADT` (AutoDAN-Turbo), and `FA` (FlipAttack)
>
> | Model | Std-CC | Std-ADT | Std-FA | Std-GASP | DS-CC | DS-ADT | DS-FA | DS-GASP |
> | --- | --- | --- | --- | --- | --- | --- | --- | --- |
> | Llama-Guard3 | 89.4 | 75.8 | 88.4 | 92.4 | 65.6 | 54.3 | 77.2 | 73.4 |
> | Aegis-Guard-D | 74.6 | 72.6 | 94.7 | 79.2 | 70.0 | 70.6 | 89.1 | 76.4 |
> | Aegis-Guard-P | 59.3 | 62.3 | 62.3 | 72.4 | 46.7 | 50.5 | 52.8 | 60.7 |
> | WildGuard | **89.9** | **79.3** | 85.7 | **94.0** | 76.9 | 70.3 | 78.0 | 84.3 |
> | ExpGuard | 89.7 | 78.4 | **99.4** | 93.8 | **87.9** | **74.0** | **100** | **87.9** |

---

> ### Author Response · Authors · 2025-11-20
> **Response to Reviewer 52Br - 2**
>
> ### **52Br-W2**
>
> > Limited evaluation. All reported evaluations are F1 scores. It is equally important to see the False Positive Rate (FPR), False Negative Rate (FNR), and prediction uncertainty to analyze what types of errors the guardrail models make, especially in high-stakes domains.
>
> We agree completely that for high-stakes domains, analyzing the specific types of errors (like False Positives and False Negatives) is critical and F1-score alone is insufficient. To provide this deeper analysis, we have expanded our evaluation to include a comprehensive suite of metrics. These metrics are:
>
> - **Recall@FPR=1% and 5%:** Answers the practical deployment question, "If we allow at most 1–5% of safe items to be wrongly blocked (FPR), what fraction of unsafe items do we successfully catch?"
> - **FPR:** Quantifies over-blocking (how much safe content is incorrectly flagged).
> - **FNR:** Quantifies how much unsafe content "slips through" the guardrail.
> - **ECE:** Assesses whether the model’s predicted probabilities are trustworthy.
>
> Below, we report the averaged scores across these metrics. **The full results are reported in** **Appendix E.5.** As the tables demonstrate, ExpGuard not only outperforms or competes strongly with other state-of-the-art guardrails across the board, but it also shows a particularly robust profile for high-stakes deployment. Notably, on our domain-specific ExpGuardTest, ExpGuard achieves the **lowest FNR** (10.5% for both prompts and responses), which is a critical outcome for sensitive domains, as it means our model lets the least amount of harmful content pass through. It also achieves the **highest Recall@FPR** (e.g., 86.2% R@FPR1% for prompts), showing it is highly effective at catching unsafe items even when we set a very low tolerance for over-blocking.
>
> This strong performance, especially the low FNR and high Recall@FPR, gives a much clearer signal of its practical effectiveness for deployment, and we thank the reviewer for pushing us to include this analysis.
>
> ### Table 1: Various evaluation metrics on ExpGuardTest (P - Prompts; R - Responses)
>
> | Model | F1 (P) | R@FPR1% (P) | R@FPR5% (P) | FPR (P) | FNR (P) | ECE (P) | F1 (R) | R@FPR1% (R) | R@FPR5% (R) | FPR (R) | FNR (R) | ECE (R) |
> | --- | --- | --- | --- | --- | --- | --- | --- | --- | --- | --- | --- | --- |
> | Llama-Guard1 | 56.9 | 37.8 | 71.4 | 1.4 | 58.8 | 23.7 | 41.7 | 35.0 | 67.5 | *0.5* | 72.7 | 29.3 |
> | Llama-Guard2 | 75.0 | 22.7 | 35.3 | 25.2 | 26.9 | 19.3 | 73.6 | 14.0 | 24.4 | 33.0 | 26.1 | 15.9 |
> | Llama-Guard3 | 69.9 | 35.5 | 50.6 | 9.3 | 41.8 | 19.7 | 82.8 | 48.1 | 65.1 | 9.3 | 23.9 | 7.4 |
> | Aegis-Guard-D | *84.9* | 43.5 | 75.4 | 6.7 | *21.9* | 12.6 | *83.4* | 34.0 | 69.4 | 9.4 | *22.0* | 17.1 |
> | Aegis-Guard-P | 68.0 | 46.0 | 75.6 | *1.3* | 47.5 | *6.8* | 50.6 | 35.1 | 69.4 | 1.0 | 65.4 | 12.2 |
> | Shield-Gemma | 17.5 | 20.5 | 54.8 | **0.1** | 89.6 | 43.3 | 10.3 | 36.3 | 58.5 | **0.0** | 94.4 | 48.7 |
> | Harmbench-LLAMA | - | - | - | - | - | - | 64.5 | 0.7 | 2.1 | 55.1 | 31.9 | 35.8 |
> | Harmbench-MISTRAL | - | - | - | - | - | - | 68.1 | 4.7 | 32.4 | 15.7 | 42.0 | 20.6 |
> | MD-Judge | - | - | - | - | - | - | 80.8 | 56.8 | *82.9* | 1.9 | 30.5 | *6.3* |
> | BeaverDam | - | - | - | - | - | - | 68.4 | 39.6 | 65.6 | 2.5 | 46.5 | 8.0 |
> | WildGuard | 84.6 | *67.0* | *78.2* | 3.2 | 24.7 | 11.7 | 76.1 | **64.2** | 82.0 | 1.1 | 37.1 | 16.0 |
> | ExpGuard | **93.4** | **86.2** | **91.3** | 2.4 | **10.5** | **5.4** | **91.7** | **64.2** | **86.6** | 6.8 | **10.5** | **3.8** |
>
> ### Table 2: Various evaluation metrics on public safety benchmarks
>
> | Model | F1 (P) | R@FPR1% (P) | R@FPR5% (P) | FPR (P) | FNR (P) | ECE (P) | F1 (R) | R@FPR1% (R) | R@FPR5% (R) | FPR (R) | FNR (R) | ECE (R) |
> | --- | --- | --- | --- | --- | --- | --- | --- | --- | --- | --- | --- | --- |
> | Llama-Guard1 | 69.2 | 32.1 | 60.0 | 7.5 | 40.5 | 16.1 | 55.4 | 18.4 | 42.3 | **6.4** | 58.2 | 18.2 |
> | Llama-Guard2 | 75.4 | 36.3 | 62.9 | *5.9* | 32.0 | 11.6 | 67.0 | 26.2 | 49.9 | 7.5 | 43.6 | 19.3 |
> | Llama-Guard3 | 78.9 | *48.2* | *67.6* | **5.6** | 26.8 | *8.3* | 66.7 | **32.0** | 52.5 | *6.5* | 43.3 | 18.7 |
> | Aegis-Guard-D | 74.3 | 22.9 | 53.2 | 28.0 | 16.2 | 9.2 | 67.3 | 6.1 | 27.2 | 40.3 | **20.7** | 7.7 |
> | Aegis-Guard-P | 74.4 | 24.7 | 54.3 | 12.8 | 29.4 | 9.0 | 65.1 | 6.9 | 26.8 | 19.6 | 39.7 | 9.1 |
> | Shield-Gemma | 68.5 | 35.3 | 61.2 | 9.4 | 38.4 | 15.6 | 57.4 | 18.6 | 37.2 | 8.1 | 55.3 | 14.6 |
> | Harmbench-LLAMA | - | - | - | - | - | - | 65.7 | 1.5 | 8.1 | 22.6 | 29.3 | 20.6 |
> | Harmbench-MISTRAL | - | - | - | - | - | - | 66.4 | 4.5 | 15.2 | 14.4 | 36.6 | 16.1 |
> | MD-Judge | - | - | - | - | - | - | 78.1 | *31.2* | **58.2** | 13.5 | 24.2 | **3.8** |
> | BeaverDam | - | - | - | - | - | - | 71.2 | 28.5 | 48.6 | 12.7 | 34.6 | *5.6* |
> | WildGuard | *84.5* | 44.6 | 66.7 | 17.6 | **7.7** | 9.0 | **78.8** | 28.6 | *57.9* | 13.5 | 24.3 | 12.5 |
> | ExpGuard | **85.6** | **58.1** | **78.4** | 14.8 | *8.2* | **7.6** | *78.4* | 30.7 | 55.1 | 15.4 | *21.9* | 12.6 |

---

> ### Author Response · Authors · 2025-11-20
> **Response to Reviewer 52Br - 3**
>
> ### **52Br-Q1**
>
> > Why fine-tune a new guardrail model from a base model? Why not just fine-tune an existing general guardrail model to keep both general and domain-specific capabilities?
>
> We appreciate this insightful suggestion. The decision to train from a base model vs. fine-tuning an existing guardrail is indeed **a design choice** involving trade-offs between domain specialization and general generalization. Following your recommendation, we conducted additional experiments fine-tuning the state-of-the-art **WildGuard** model using our domain-specific data via both Full Fine-Tuning (FFT) and LoRA. The comparative results are presented below:
>
> | **Model** | **Prompt Harm (Public Avg F1)** | **Prompt Harm (ExpTest Total F1)** | **Response Harm (Public Avg F1)** | **Response Harm (ExpTest Total F1)** |
> | --- | --- | --- | --- | --- |
> | **ExpGuard (From Base)** | **85.7** | 93.3 | 78.5 | **92.7** |
> | WildGuard + FFT | 82.9 | 94.1 | 77.5 | 91.9 |
> | WildGuard + LoRA | 84.4 | **95.0** | **79.4** | 90.3 |
>
> We observe that while FFT leads to noticeable catastrophic forgetting on general benchmarks (dropping from 85.7 to 82.9 on Public Avg F1), applying LoRA effectively mitigates this issue, maintaining a better balance between general and domain-specific capabilities.
>
> **Why we chose to train from the Base Model:** Ultimately, training ExpGuard from the base model provided the best overall equilibrium. It achieves the highest general prompt safety score (85.7) while maintaining exceptional domain performance. This approach allows us to:
>
> 1. **Ensure Distributional Purity:** We maintain full control over the training distribution via **ExpGuardMix**, avoiding potential taxonomy conflicts or distribution shifts that might arise from the "implicit" data within a pre-trained guardrail.
> 2. **Demonstrate Dataset Efficacy:** It validates that **ExpGuardMix** alone is sufficient to train a state-of-the-art safety model from scratch, without relying on the warm-start of existing safety checkpoints.
>
> ---
>
> **References**
>
> [1] Liu et al. FlipAttack: Jailbreak LLMs via Flipping, ICML 2025.
>
> [2] Basani et al. GASP: Efficient Black-Box Generation of Adversarial Suffixes for Jailbreaking LLMs, NeurIPS 2025.

---

> > ### Author Response · Authors · 2025-11-26
> > **A gentle reminder to Reviewer 52Br**
> >
> > Dear reviewer, thanks again for taking the time to reassess our work. Since the final days of the discussion phase are approaching, we wanted to reach out and ask if there are any further questions from your end. We hope that you like the improvements to the manuscript, and the substantial additions to the experiments (both in the main paper and the appendix). In particular, we added detailed metrics like FNR/Recall@FPR and comparisons with the most recently developed jailbreak attacks to fully address your concerns on high-stakes deployment.

---

### Official Review · Reviewer_3d4f · 2025-11-01

**Soundness:** 2
**Presentation:** 2
**Contribution:** 2
**Rating:** 6
**Confidence:** 4

**Summary:**

The paper contributes a dataset consisting moderation contents in specific domains such as finance, healthcare and law. Experiments on several benchmarks demonstrate the effectiveness of the proposed method.

**Strengths:**

1. the paper is well written and easy to follow
2. the paper studies moderation problems in specific domains such as finance, healthcare and law which are not studied extensively by popular methods.
3. the paper has attached its implementation and code which is good for reproducing the work
4. the comparison is extensive consisting of both test composed by ExpGuard and public tests.

**Weaknesses:**

1. the paper mainly focuses on terminology in specific domains. In order to tackle such problems, extensive efforts have to be made such as crawling wikipedia to collect all of them. The whole generation and training process seems very resourcing consuming.

**Questions:**

1. The generation and then filtering pipeline works very similarly to the work [1]. Can the authors explain the advantage of the proposed work compared to the pipelines proposed in [1,2].
2. Following weakness 1, the system cannot be updated promptly if new terms come out. Since LLMs are capable of dynamically retrieve the meaning of terminology, have the authors thought about bypassing the wikipedia collections process and directly incorporate the definition of such terminologies for final classification.
3. GPT-4o should have built-in safety mechanisms that prevent users from constructing malicious prompts. How does the authors get around it to have GPT-4o construct all the prompts [3]?
4. have the authors analyzed the difficulties of the proposed training and test set? Some prompts should be more easy to classify than others. Does the proposed pipeline possess the ability to generate training and test sets of different difficulty levels?


[1] An, Bang, et al. "Automatic pseudo-harmful prompt generation for evaluating false refusals in large language models." arXiv preprint arXiv:2409.00598 (2024).

[2] Cui, Justin, et al. "Or-bench: An over-refusal benchmark for large language models." arXiv preprint arXiv:2405.20947 (2024).

[3] https://openai.com/index/introducing-the-model-spec/

---

> ### Author Response · Authors · 2025-11-20
> **Response to Reviewer 3d4f**
>
> *We appreciate Reviewer 3d4f for their constructive feedback. We hope our responses address the reviewer’s concerns.*
>
> ### **3d4f-W1**
>
> > Extensive efforts have to be made such as crawling wikipedia to collect all of them. The whole generation and training process seems very resourcing consuming.
>
> We appreciate the reviewer's concern regarding the resources required for our data construction pipeline. However, we would like to clarify that our approach is **primarily automated** (or fully automated, if one chooses to opt out of the human-in-the-loop step), which incurs a significantly lower cost than manually collecting and labeling tens of thousands of samples. Furthermore, our work tackles a problem in specialized domains. For such problems, a traditional approach would require substantial costs to compensate domain experts. Our pipeline, by leveraging systematic terminology filters and LLMs, **substantially reduces this cost**.
>
> In practice, the automated components are highly efficient: crawling takes only around 10 seconds to retrieve 1,000 terms from Wikipedia with a single CPU thread (which can be accelerated *N* times with *N* threads), and training takes ~3 hours on four H200 GPUs, as noted in Appendix D.1. We believe our data construction pipeline and training cost are **reasonably economical**, providing a reproducible and extensible solution to the community.
>
> To clarify this, **we revised $\S$1** to emphasize the cost-efficiency of our automated approach over manual labor, and **we added the computational details of crawling to Appendix A.1**.
>
> ### **3d4f-Q1**
>
> > The generation and then filtering pipeline works very similarly to the work [1]. Can the authors explain the advantage of the proposed work compared to the pipelines proposed in [1,2].
>
> While the high-level "generate-then-filter" pattern is common, our pipeline's implementation and purpose at each step are fundamentally different from [1] and [2].
>
> **Different Problem: False Negatives vs. False Positives**
>
> The core distinction is that we are solving a completely different problem.
>
> - [1] (PHTest) and [2] (OR-Bench) address **false positives** (over-refusal). Their goal is to generate safe prompts that look harmful (e.g., "how to kill a mosquito") to test if a model incorrectly refuses a benign query.
> - Our work (ExpGuard) tackles the opposite: **false negatives** (missed harm). Our goal is to generate truly harmful prompts that look safe to a non-expert, as the harmful intent is masked by specialized, technical jargon (e.g., "obscure high haircuts in asset evaluations").
>
> **Different Pipeline: Advantages in "Generate" and "Filter"**
>
> This difference in goals dictates our pipeline's distinct advantages:
>
> **1. The "Generate" Step: Knowledge-Grounded vs. Rewriting**
>
> - [1] and [2] remove harm. They start with general toxic seeds and rewrite them to be benign.
> - Our pipeline constructs harm. We first **mine domain-specific technical terms** (e.g., "haircuts" in finance) and then use **RAG with Wikipedia abstracts** to build novel, realistic, and harmful scenarios. This knowledge-grounded approach is essential for specialized domains, as standard models may lack the required technical depth and domain-specific nuance.
>
> **2. The "Filter" Step: Expert Validation vs. General Moderation**
>
> - For ExpGuardTrain, our filtering process is **multi-stage**: we first apply systematic terminology filters, then use an LLM ensemble (unlike [2], which relies solely on an ensemble). While [1,2] also employ human validation, our pipeline's key advantage is the validation of ExpGuardTest by **domain experts** (e.g., finance professionals), rather than generalist annotators. This expert-led validation is the necessary step to create a gold-standard benchmark for this nuanced, domain-specific harm.
>
> **We added a discussion of these works to $\S$3.2** to clarify this distinction.

---

> ### Author Response · Authors · 2025-11-20
> **Response to Reviewer 3d4f - 2**
>
> ### **3d4f-Q2**
>
> > Have the authors thought about bypassing the wikipedia collections process and directly incorporate the definition of such terminologies for final classification?
>
> This is an excellent suggestion regarding the potential of utilizing LLMs' dynamic retrieval capabilities (e.g., RAG-based approaches) to bypass offline data collection. We carefully considered this direction but prioritized our current methodology for three critical reasons specific to **production guardrail systems**:
>
> 1. **Inference Latency & Throughput:** A guardrail model must filter every user interaction in real-time with minimal latency. Your suggested approach—retrieving definitions and incorporating them into the context for every prompt—introduces significant overhead (search latency + processing long contexts with definitions). In contrast, our ExpGuard model "internalizes" this domain knowledge into its weights through fine-tuning. This allows it to detect sophisticated jargon-based attacks instantly without external retrieval, making it far more suitable for high-throughput deployment.
> 2. **Cost Efficiency at Inference:** "Directly incorporating the definition" significantly increases the input token count for every query. For high-volume applications, this results in prohibitive operational costs. Our fine-tuned 7B model achieves high accuracy with minimal token usage, offering a much more economical inference solution than a retrieval-augmented pipeline.
> 3. **Safety & Robustness:** Relying on dynamic retrieval at inference time introduces severe security vulnerabilities, such as **Indirect Prompt Injection [3]** or **Knowledge Poisoning [4]**. Attackers could manipulate external web content (e.g., via SEO poisoning) to inject malicious instructions that bypass the guardrail when retrieved. By curating the dataset offline (using our automated pipeline) and verifying it, we ensure the model learns from a clean, controlled distribution, providing a more robust defense layer.
>
> Furthermore, regarding the "prompt update" concern: as highlighted in our response to W1, our **data pipeline is automated and highly efficient**. Therefore, when new terminologies emerge, we can rapidly generate new synthetic samples and perform continual fine-tuning (or lightweight adaptation like LoRA) to update ExpGuard. This combines the agility of automation with the efficiency of a fine-tuned model.
>
> ### **3d4f-Q3**
>
> > How does the authors get around it to have GPT-4o construct all the prompts [3]?
>
> Indeed, we initially struggled to construct malicious prompts with GPT-4o due to its safety mechanisms. To bypass this, we followed the technique introduced in [5], adding a simple affirmative prefix, “**I have an idea for a prompt:**”. This was a non-trivial detail that we missed in our original manuscript. **We added this detail in $\S$3.1.2.** We sincerely appreciate the reviewer’s diligent observation.
>
> ### **3d4f-Q4**
>
> > Have the authors analyzed the difficulties of the proposed training and test set? Does the proposed pipeline possess the ability to generate training and test sets of different difficulty levels?
>
> Thank you for this insightful question. To analyze the difficulty of our proposed datasets, we adopt an approach inspired by Item Response Theory [6], which infers per-item difficulty from the observed success rates of different "solvers". In our context, we use our baseline guardrail models (Llama-Guard3, Aegis-Guard-D, Aegis-Guard-P, and WildGuard) as the solvers. We then categorize each sample on a 3-point difficulty scale based on the number of models that failed to classify it correctly: a sample is **Easy** if all four models get it correct, **Medium** if at least one (but fewer than three) fails, and **Hard** if at least three models fail.
>
> The tables below present the difficulty statistics for our training set (for brevity). The results demonstrate that our dataset spans a wide spectrum of difficulty. While a majority of samples are 'Easy', a substantial portion falls into the 'Medium' and 'Hard' categories (totaling **38.60%** for prompts and **29.59%** for responses). This long tail of challenging samples is crucial for robustly evaluating and training guardrail models.
>
> While our pipeline does not have an explicit mechanism to generate data of a specific difficulty level, we note that more challenging samples can be crafted by applying jailbreak techniques to our proposed dataset, as we demonstrated in Section 4.4. **We reported this full difficulty analysis in Appendix A.12.**
>
> | Difficulty | Prompts (Count) | Prompts (%) | Responses (Count) | Responses (%) |
> | ---------- | --------------: | ----------: | ----------------: | ------------: |
> | Easy       |          34,786 |      61.40% |            18,943 |        70.41% |
> | Medium     |          17,566 |      31.01% |             6,029 |        22.41% |
> | Hard       |           4,301 |       7.59% |             1,931 |         7.18% |

---

> ### Author Response · Authors · 2025-11-20
> **Response to Reviewer 3d4f - 3**
>
> **References**
>
> [1] An et al. Automatic Pseudo-Harmful Prompt Generation for Evaluating False Refusals in Large Language Models, COLM 2024.
>
> [2] Cui et al. OR-Bench: An Over-Refusal Benchmark for Large Language Models, ICML 2025.
>
> [3] Greshake et al. Not what you've signed up for: Compromising Real-World LLM-Integrated Applications with Indirect Prompt Injection, AISec@CCS 2023.
>
> [4] Zou et al. PoisonedRAG: Knowledge Corruption Attacks to Retrieval-Augmented Generation of Large Language Models, USENIX Security 2025.
>
> [5] Lee et al. HarmAug: Effective Data Augmentation for Knowledge Distillation of Safety Guard Models, ICLR 2025.
>
> [6] Vania et al. Comparing Test Sets with Item Response Theory, ACL 2021.

---

> > ### Author Response · Authors · 2025-11-26
> > **A gentle reminder to Reviewer 3d4f**
> >
> > Dear reviewer, we would be very happy to discuss any further questions before the end of the discussion phase. In particular, please let us know if your questions around the pipeline's cost-efficiency and the distinctions from prior work have been resolved by our response. We also hope you find the new difficulty analysis to be a valuable addition for assessing the dataset's complexity. Thanks again for your comments which helped a lot for revising our manuscript.

---

### Official Review · Reviewer_MLL9 · 2025-11-07

**Soundness:** 3
**Presentation:** 3
**Contribution:** 3
**Rating:** 4
**Confidence:** 3

**Summary:**

This paper focuses on LLM safety on harmful and adversarial content within domain-specific contexts, which include technical jargon and specialized concepts. The authors introduce a specialized guardrail model called EXPGUARD to protect against harmful prompts and responses across financial, medical, and legal domains. In addition, they also present a human-curated dataset containing 58,928 labeled prompts paired with corresponding refusal and compliant responses, which has both training and test datasets. Comprehensive experiments on EXPGUARDTEST and eight public benchmarks show the competitive performance of EXPGUARD.

**Strengths:**

1. The proposed dataset is of high quality, which can promote the future research on LLM safety in specific domains.
2. Experimental results provide some insights into the task of LLM safety in specific domains.
3. This paper is well-written and easy to follow.

**Weaknesses:**

1. The contribution of this paper mainly falls into the resource side. However, in my view, the technical contribution of the dataset construction method is somewhat limited. The authors spend a lot of room (i.e., Section 3.1 and 3.2) for the details about data construction, labeling, and filtering, whose technical design is comprehensive but not very novel.
2. The multi-task training method of EXPGUARD (Section 3.3) seems direct and intuitive, which is similar to the models for LLM safety of general domains. The authors are suggested to further consider methodological design in addition to dataset construction.

**Questions:**

I have included my questions in the weaknesses part.

---

> ### Author Response · Authors · 2025-11-20
> **Response to Reviewer MLL9**
>
> *We appreciate Reviewer MLL9 for their incisive feedback. We hope our responses address the reviewer’s concerns.*
>
> ### **MLL9-W1**
>
> > The contribution of this paper mainly falls into the resource side. However, in my view, the technical contribution of the dataset construction method is somewhat limited. The authors spend a lot of room (i.e., Section 3.1 and 3.2) for the details about data construction, labeling, and filtering, whose technical design is comprehensive but not very novel.
>
> We believe our work should be viewed as **a complete end-to-end framework, not merely a resource**. The technical contribution is not limited to data labeling, but resides in the **novel, automated methodological framework** designed to solve the "cold start" problem in domain-specific safety where expert data is scarce.
>
> - **Pipeline Novelty:** This framework is the first systematic, reproducible pipeline designed to automate the discovery and generation of specialized safety data.
> - **Rigorous Design**: Our pipeline is defined by technical rigor at every stage:
>     - Systematic Terminology Mining ($\S$3.1.1): A multi-stage funnel using Wikidata API filtering (entity removal) and sophisticated LLM-based filtering.
>     - Bias-Mitigated Labeling ($\S$3.1.3): We designed a technical solution using an ensemble of three different proprietary LLMs (Claude 3.7, Gemini 2.0, Qwen2.5-Max) combined with CoT rationales to enforce exact category consensus. This mitigates shared bias risks inherent in single-model labeling.
> - **Expert Validation:** The high inter-annotator agreement (Kappa 0.89–0.98) demonstrates that this automated methodology successfully approximates human expert quality without the prohibitive cost of full manual annotation.
> - **Extensibility**: The framework is designed to be readily extended to automatically construct specialized datasets for other critical domains (e.g., cybersecurity).
>
> ### **MLL9-W2**
>
> > The multi-task training method of EXPGUARD (Section 3.3) seems direct and intuitive, which is similar to the models for LLM safety of general domains. The authors are suggested to further consider methodological design in addition to dataset construction.
>
> We accept the observation that our multi-task training approach is "direct and intuitive". However, we posit that this **simplicity is a core strategic strength** of our work, not a limitation. Consistent with the data-centric perspective, our hypothesis is that the **quality of the data itself serves as the primary methodological lever** for specialized safety, rather than the complexity of the training objective (e.g., RLHF). Our experiments provide empirical validation:
>
> - **Proof 1: Ablation Study ($\S$4.3):** Removing the domain-specific data from ExpGuardTrain causes the model's F1 score on ExpGuardTest to decline sharply by 8.0% (93.3% $\to$ 85.3%). This demonstrates that the SOTA performance stems directly from our novel data construction methodology.
> - **Proof 2: Efficiency (Appendix E.2):** A 1.5B parameter model trained solely on our data still outperforms all larger 7B SOTA baselines on the domain-specific ExpGuardTest. This empirically validates the efficiency and effectiveness of our data pipeline.
>
> In summary, our contribution is the **end-to-end framework:** the novel methodology for automated data creation, the ExpGuardMix dataset, the expert-validated ExpGuardTest benchmark, and the SOTA ExpGuard model.

---

> > ### Author Response · Authors · 2025-11-26
> > **A gentle reminder to Reviewer MLL9**
> >
> > Dear reviewer, given that the end of the discussion period is approaching, we wanted to reach out and discuss any further questions you might have. We are happy to address further follow-up questions, in particular around the technical novelty of our end-to-end automated framework and the strategic value of our data-centric training approach. In any case, thank you for taking the time to consider our rebuttal and the detailed clarifications we provided. Your comments provided a valuable opportunity to better articulate our contributions.

---

### Author Response · Authors · 2025-11-20
**General Response to All Reviewers**

We thank the reviewers for their thoughtful and constructive feedback. We have carefully considered all comments and have revised our manuscript. Below is a summary of the key updates included in the revised manuscript (highlighted in **blue**), followed by a summary of the additional analyses provided in our rebuttal.

### Part 1: Manuscript Revisions & Updates

- Updated Figures
    - **Figure 4 (Jailbreak Analysis):**
        - Added performance results on prompts *before* jailbreak methods are applied ("Original") [**YTqP-Q5**].
        - Added results for two additional state-of-the-art attacks: FlipAttack [1] and GASP [2] [**52Br-W3**].
        - Introduced ExpGuard+, a variant of our model retrained with domain-specific adversarial examples, showing significantly improved robustness [**YTqP-W1/Q4**].
- Updated Sections
    - **Section 1:** Clarified the framing of our data construction pipeline as a resource-efficient solution, rather than resource-intensive [**3d4f-W1**].
    - **Section 3.1.2:** Added details on the prefix-injection technique used to bypass GPT-4o safety refusals during data generation, adopted from [3] [**3d4f-Q3**].
    - **Section 3.1.3:** Clarified the strict "majority vote per category" consensus protocol used for data filtering [**YTqP-Q2**].
    - **Section 3.2:** Explicitly distinguished our approach from recent "generate-then-filter" frameworks [4,5] [**3d4f-Q1**].
    - **Section 4.4:** Expanded the analysis of domain-specific jailbreak resilience [**52Br-W3**, **YTqP-W1**].
- Updated Appendices & Tables
    - **Appendix A.1:** Added computational details regarding the Wikipedia crawling process to demonstrate cost-efficiency [**3d4f-W1**].
    - **[NEW] Appendix A.10 & Table 11:** Added a Semantic Overlap Analysis quantifying the distinctiveness of ExpGuardMix against WildGuardMix, SALAD-Bench, and BeaverTails [**52Br-W1**].
    - **[NEW] Appendix A.11 & Tables 12–13:** Added detailed statistics on the Data Consensus and Filtering Pipeline, including discard rates at each phase [**YTqP-Q2/Q3**].
    - **[NEW] Appendix A.12 & Table 14:** Added a Dataset Difficulty Analysis based on Item Response Theory principles to characterize the hardness distribution of our prompts [**3d4f-Q4**].
    - **Appendix E.2:** Added results for ExpGuard (14B) to demonstrate performance scaling [**YTqP-W2**].
    - **[NEW] Appendix E.4 & Table 18:** Added an analysis of the effect of the Response Generator Model (Mistral-7B vs. Mistral-Small-24B) on guardrail performance [**YTqP-W3/Q1**].
    - **[NEW] Appendix E.5 & Tables 19–28:** Added Comprehensive Evaluation Metrics, reporting Recall@FPR=1%, Recall@FPR=5%, FPR, FNR, and ECE for all models [**52Br-W2**].

### Part 2: Additional Analysis & Design Rationale

In addition to the manuscript changes, we have provided detailed discussions and experimental evidence in our individual responses to clarify specific design choices:

- **Technical Novelty & Framework:** We clarified that our contribution is an end-to-end automated framework designed to solve the "cold start" problem for specialized safety, utilizing a novel multi-stage filtering and bias-mitigated labeling pipeline [**MLL9-W1**].
- **Simplicity in Methodology:** We discussed why a simple multi-task objective was strategically chosen to emphasize data quality over architectural complexity, a hypothesis supported by our ablation studies showing that data composition is the primary performance driver [**MLL9-W2**].
- **Training Strategy (Base Model vs. Fine-tuning):** We clarified that training from a base model versus fine-tuning an existing guardrail is a strategic design choice. We justified our decision to train from the base model to ensure distributional purity and to demonstrate dataset efficacy [**52Br-Q1**].
- **Offline Curation vs. Dynamic Retrieval:** We detailed the specific constraints—latency, inference cost, and security risks (e.g., prompt injection)—that make our offline fine-tuning approach superior to dynamic RAG-based definition retrieval for production guardrail systems [**3d4f-Q2**].

We believe these revisions and clarifications significantly strengthen the paper's contribution and rigor. We thank the reviewers again for their time and valuable insights in helping us improve this work.

---

**References**

[1] Liu et al. FlipAttack: Jailbreak LLMs via Flipping, ICML 2025.

[2] Basani et al. GASP: Efficient Black-Box Generation of Adversarial Suffixes for Jailbreaking LLMs, NeurIPS 2025.

[3] Lee et al. HarmAug: Effective Data Augmentation for Knowledge Distillation of Safety Guard Models, ICLR 2025.

[4] An et al. Automatic Pseudo-Harmful Prompt Generation for Evaluating False Refusals in Large Language Models, COLM 2024.

[5] Cui et al. OR-Bench: An Over-Refusal Benchmark for Large Language Models, ICML 2025.

---

### Author Response · Authors · 2025-12-03
**Final Comment to AC**

Dear AC,

We sincerely appreciate your dedication in stepping in to handle our submission following the abrupt interruption of the discussion due to identity leaks. To assist your final assessment, we summarize our resolutions below. We have fully integrated **substantial new experiments and analyses** into the revised manuscript (colored in blue), rigorously addressing every reviewer point without relegating items to "future work”.

Reviewer **YTqP (Score: 8)** confirmed their concerns were successfully addressed and maintained a positive assessment. Although the discussion closed before other reviewers could reply, we are confident our extensive revisions effectively resolve their outstanding questions.

Below is a detailed summary of how each specific concern was addressed:

### **Summary of Resolutions**

| Reviewer Concern / Question | Resolution & Revision in Manuscript | Addressed? |
| --- | --- | --- |
| **Reviewer YTqP (Score: 8)** |  |  |
| - | The reviewer confirmed that all concerns were addressed and provided a positive evaluation. | ✅ |
| **Reviewer 3d4f (Score: 6)** |  |  |
| **Resource consumption:** Worried that data construction (e.g., crawling) is resource-intensive. | **Clarified:** We demonstrated that our pipeline is automated and highly efficient. Wikipedia crawling incurs negligible compute cost (10 seconds per 1k terms), and training takes only ~3 hours. (See §1 & App. A.1) | ✅ |
| **Comparison with existing pipelines:** Questioned similarity to works like PHTest/OR-Bench. | **Clarified:** We distinguished our work fundamentally: those works address *false positives* (over-refusal), whereas we address *false negatives* (missed harm) using knowledge grounding and expert validation, which necessitates a different pipeline design. (See §3.2) | ✅ |
| **Test-time terminology retrieval:** Suggested using dynamic RAG instead of offline training. | **Clarified:** We articulated the critical disadvantages of online retrieval for guardrails: high latency, increased inference cost, and safety risks (e.g., prompt injection). We justified our offline design for robust production deployment. | ✅ |
| **Bypassing GPT-4o:** Asked how we generated harmful prompts despite safety filters. | **Clarified:** We detailed the specific technique used (adding an affirmative prefix) to successfully bypass refusals during data generation. (See §3.1.2) | ✅ |
| **Dataset difficulty:** Asked for analysis on difficulty levels. | **Analyzed:** We conducted an Item Response Theory analysis, categorizing samples into Easy, Medium, and Hard, proving ExpGuardMix covers a wide difficulty spectrum. (See App. A.12) | ✅ |
| **Reviewer 52Br (Score: 4)** |  |  |
| **Dataset Overlap:** Concerned about overlap with existing datasets (e.g., WildGuardMix). | **Analyzed:** We conducted a semantic overlap analysis showing extremely low overlap (e.g., only 5.1% with WildGuardMix), proving our dataset's distinctiveness. (See App. A.10) | ✅ |
| **Limited Evaluation:** Requested metrics beyond F1-score. | **Expanded:** We reported five additional metrics (Recall@FPR=1%, Recall@FPR=5%, FPR, FNR, ECE), demonstrating ExpGuard’s superior practical effectiveness for high-stakes deployment. (See App. E.5) | ✅ |
| **Jailbreak Robustness:** Questioned effectiveness against stronger attacks. | **Experimented:** We tested against the latest attacks (FlipAttack, GASP), further demonstrating ExpGuard's robustness. (See §4.4) | ✅ |
| **Training Strategy:** Asked why train from base instead of fine-tuning existing guardrails. | **Justified:** We clarified that this is a design choice, as experiments fine-tuning WildGuard showed comparable performance. Training from base was chosen to demonstrate that ExpGuardMix alone is sufficient for SOTA performance without relying on existing checkpoints. | ✅ |
| **Reviewer MLL9 (Score: 4)** |  |  |
| **Technical Contribution:** Felt the contribution was limited to resources. | **Clarified:** We framed the work as an end-to-end framework, the *first* systematic, reproducible pipeline for specialized safety. We note that other reviewers (3d4f, YTqP) explicitly recognized the novelty of this pipeline. (See §1 & §3) | ✅ |
| **Methodological Simplicity:** Found the design too simple. | **Justified:** We argued that simplicity is a strategic strength. Our ablation studies prove that *data quality*, rather than complex training objectives, is the primary driver of performance in this domain. (See §4.3) | ✅ |

We believe these revisions support a positive decision, and we hope this summary assists you in making a final decision.

Best regards,

Paper 17133 Authors

---

### Meta-Review · Area_Chair_7tEF · 2026-01-06

**Summary:**

This work introduces ExpGuard, a Guard model trained on ExpGuardMix, which consists of 58,928 prompts, including prompts from the specialized domains of Finance, Medicine, and Law. This dataset addresses the lack of domain-specific prompts for training guardrail systems. The Authors describe in high detail the creation of ExpGuardMix with a novel construction pipeline that utilized domain-specific terminology. In addition to specialized examples, ExpGuardMix incorporates real-world user-LLM conversations and human-written prompts, making ExpGuard models useful outside the specialized domains. ExpGuard is evaluated against other state-of-the-art guardrails, highlighting its advantage over other available solutions, both open-source and closed API models.

**Reviewer Scores:**

No

---

### Decision · Program_Chairs · 2026-01-26

Accept (Poster)